# Carbon Stock and Change Rate under Different Grazing Management Practices in Semiarid Pastoral Ecosystem of Eastern Ethiopia

**Haftay Hailu Gebremedhn** [1],*, **Tessema Zewdu Kelkay** [2], **Yayanshet Tesfay** [3], **Samuel Tuffa** [4], **Sintayehu Workeneh Dejene** [1], **Sylvanus Mensah** [5], **Adam John Mears Devenish** [6] and **Anthony Egeru** [7]

1   African Center of Excellence for Climate-Smart Agriculture and Biodiversity Conservation, Haramaya University, Dire Dawa P.O. Box 138, Ethiopia; sintekal@gmail.com
2   College of Agriculture and Environmental Sciences, Debark University, Debark P.O. Box 90, Ethiopia; tessemaz@yahoo.com
3   College of Dryland Agriculture and Natural Resources, Mekele University, Mekelle P.O. Box 23, Ethiopia; yayneshet_tesfay@yahoo.com
4   Oromia Agricultural Research Institute, Addis Ababa P.O. Box 81265, Ethiopia; satukada@gmail.com
5   Laboratoire de Biomathématiques et d'Estimations Forestières, Université d'Abomey Calavi, Cotonou 04 BP 1525, Benin; m.sylvanus@ruforum.org
6   Department of Life Sciences, Imperial College London, London SW7 2AZ, UK; a.devenish@imperial.ac.uk
7   Training and Community Development, Regional Universities Forum for Capacity Building in Agriculture (RUFORUM), Kampala P.O. Box 16811, Uganda; a.egeru@ruforum.org
*   Correspondence: hailuhft418@gmail.com

**Abstract:** Grazing management strategies tend to have different effects on rangeland plant production. Changes in grazing management can, therefore, affect the carbon stock potential of rangelands. Despite rangeland ecosystems being important global sinks for carbon, we know relatively little about the effect of traditional grazing management practices on their potential to store carbon. In this study, we evaluated the carbon stock and change rate of rangelands using three traditional grazing management practices in the semiarid pastoral ecosystem of eastern Ethiopia. By comparing data on vegetation and soil carbon stocks, we found that there was a strong significant difference ($p < 0.001$) between these different management practices. In particular, the establishment of enclosures was associated with an annual increase in carbon stocks of soil (3%) and woody (11.9%) and herbaceous (57.6%) biomass, when compared to communal open lands. Both enclosure and browsing management practices were found to have the highest levels of soil organic carbon stocks, differing only in terms of the amount of woody and herbaceous biomass. Thus, modest changes in traditional grazing management practices can play an important role in carbon storage and sequestration. Further research is required on a wider range of traditional pastoral management practices across space and time, as understanding these processes is key to combating global climate change.

**Keywords:** open grazing; enclosure; browsing area; climate change mitigation; rangeland ecosystem services; carbon stock

## 1. Introduction

Carbon dioxide ($CO_2$) gas is a major contributor to global climate change. While the greenhouse gas effect is important for human survival, anthropogenic effects are leading to rapid increase of $CO_2$ gas in the atmosphere, resulting in increased global temperatures [1]. Carbon sequestration is the process of capturing and storing atmospheric $CO_2$ to reduce global climate change [2]. Sequestering carbon in plants and soil to limit the release of $CO_2$ back into the atmosphere is vital for offsetting greenhouse gas emissions [3].

Rangelands are biomes of global importance; their coverage is estimated at 54% of the world's terrestrial area and 43% of Africa's land area [4]. Accordingly, rangelands have

the potential to contribute to the mitigation of global climate change [5]. Rangelands are also thought to have as much as 30% of terrestrial carbon stocks [4]. Traditional rangeland management practices inherent to pastoralism have shaped rangeland environments for millennia and have contributed to maintain soil and plant biomass carbon stocks [6]. However, pastoralism is rarely viewed as a major opportunity in the fight against global climate change. This is because pastoral rangeland grazing areas have often been documented as highly degraded due to perceived overstocking by pastoralists. This is inducing a decline in rangeland performance in both carbon capture potential as well as other important ecosystem services [7]. Currently, there is relatively little known about how changes in grazing management affect the carbon stock potential of rangeland ecosystems [8].

Implementation of improved land management practices to build up carbon stocks in rangeland ecosystems has been shown to increase the performance of rangelands as effective carbon sinks [9,10]. For example, management activities such as (i) rotational grazing, (ii) moderate stocking rates, (iii) rehabilitating degraded rangelands with local grasses, (iv) use of prescribed fire every three or more years, and (v) thinning woody vegetation have been shown to increase rangeland carbon sequestration [10–12]. However, it is important to note that some management practices, such as increased grazing pressure (due to the overstocking of livestock), can decrease carbon stock potential [13–15]. Therefore, balancing these activities through different management practices is important for ensuring that these rangelands act globally as sinks rather than sources of $CO_2$ emissions.

Rangelands in Africa often fail to maintain their carbon stock due to degradation and overexploitation. The overutilization of plant biomass and soil erosion are contributing to the release of $CO_2$ into the atmosphere [16,17]. Maintaining and/or increasing rangeland carbon stock is, therefore, heavily dependent on a range of localized factors including land-use history, vegetation cover, grazing management practices, intensity, and soil properties (e.g., clay content, amorphous Fe and Al oxides, etc.) [2,18,19]. African rangelands' capacity to sequester carbon under different management practices varies from 0.23 to 14.33 t C ha$^{-1}$ [20]. Despite such variation, relatively little is known about the effect of traditional pastoral rangeland management practices on carbon sequestration in Africa [20,21].

The Somali Regional State in Ethiopia is one of the major pastoral ecosystems, 90% of which is rangeland [22]. Pastoralists in the Somali region have various traditional natural resource management strategies. Enclosures are one of the management strategies that is used in response to declining rangeland productivity and patchy resource distribution [23]. Enclosures in this region have been in use for a considerable time. These long-term enclosures have the potential to improve carbon sequestration; however, free-ranging of livestock on communal land remains the most common customary management practice in the region [24]. The free-ranging management strategy is deployed to make use of the heterogeneously occurring pasture and other grazing resources over the landscape.

Despite the potential for these alternative management practices to improve carbon sequestration, nomadic herders' indigenous knowledge has been often neglected by extension and research services. To date, there is little information on the variation in carbon stock across these different management practices, especially in terms of their potential for carbon sequestration and climate change mitigation. This study, therefore, assessed the potential of pastoralist management practices for carbon storage in vegetation biomass and soils. Specifically, we (i) assessed the effects of traditional grazing management systems on vegetation and soil carbon stocks and (ii) determined the relative rate of change in carbon in the enclosures, as compared to the communal open grazing areas.

## 2. Materials and Methods

### 2.1. Description of the Study Areas

This study was conducted in Jigjiga, located in the Somali Regional State (SRS) of Ethiopia (Figure 1). The Somali Region is the easternmost of the nine ethnic divisions (*Kililoch*) in Ethiopia. The latitude of Jigjiga is 9.356784, and the longitude is 42.795519.

The temperature in the Jigjiga zone is generally high all year round, and the mean minimum value is 20 °C, and the mean maximum is 35 °C [25]. The region has a bimodal rainfall pattern, with mean annual precipitation of 660 mm. The rainfall condition in the zone is characterized by low, unreliable, and uneven distribution. The soil of the Somali Regional State is dominated by weakly developed soil horizon and features of stony petro-calcic and petrogypsic phases. The dominant soil types are yermosols, xerosols, regosols, and solonchaks [26].

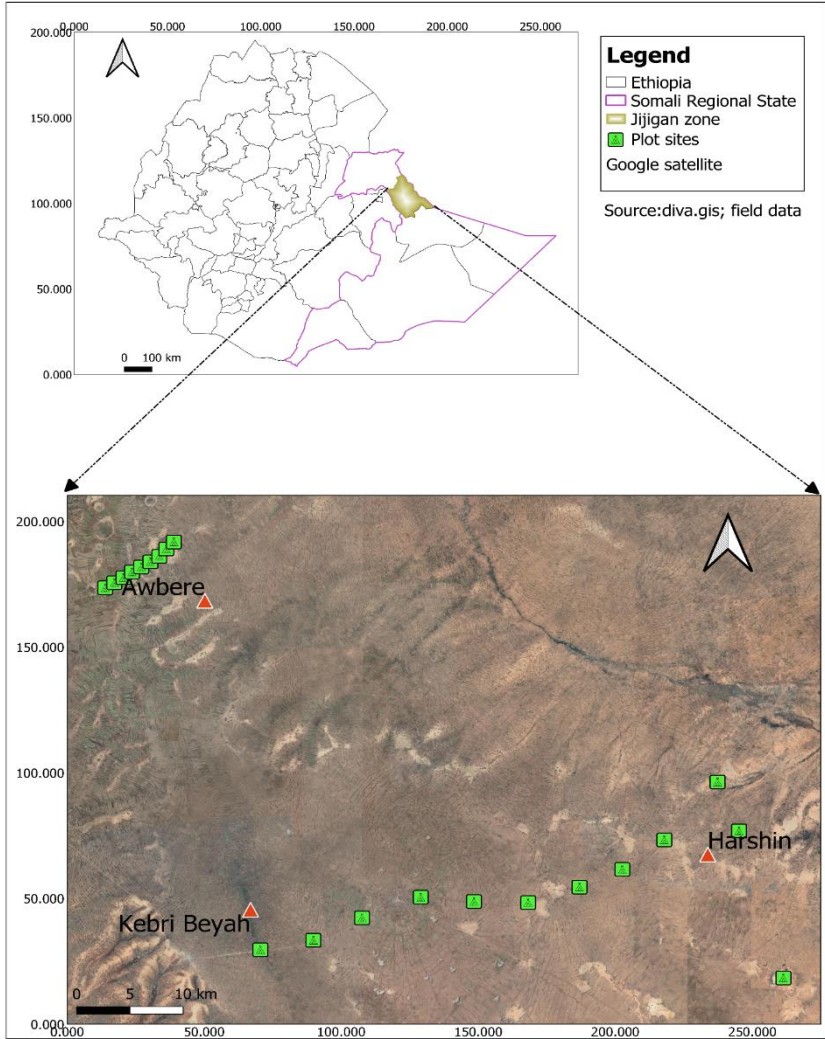

**Figure 1.** Location of the study area.

The land-use system in the zone is predominantly pastoral and agropastoral; the livestock entirely depends on natural vegetation. Pastoralists who depend on livestock are nomadic in lifestyle and use much of the land for natural fodder. Equally, there are agropastoralists in the area that cultivate small farms for subsistence purpose [27].

### 2.2. Description of the Management Systems

Pastoralists have various traditional natural resource management strategies such as management of rangeland and livestock (e.g., identifying dry and wet season grazing, herd management, controlled soil burning, proper water management systems, and weed and pest management). In this study, we selected the three main land-use management practices commonly used by Somali pastoralists.

i.    Communal open grazing: this represents the most common land-use system in the Somali rangelands. Communal open grazing land is defined as the communal

rangelands that are not privately owned but belong to the communities whose members have equal-access rights to the communal resources. This land type is characterized by open grass vegetation with scattered woody trees. The pastoralists use it for extensive livestock grazing throughout the year.

ii. Browsing land is a grazing management system that is used in response to changing environmental conditions. By splitting herd composition, it enhances climate resilience. To split the herd composition, a pastoralist uses herd diversity and ecosystem knowledge as the bases for vegetation management. Pastoralists divide grazing habitats into micro-categories based on plant cover, soil type, and ecosystem-functioning knowledge. Herd diversity also helps to ensure the optimum utilization of resources since the different domestic animals have different feed preferences. For example, camels and goats browse, while sheep and cattle graze. This maximizes the utilization of the available fodder. The grazing management system that is commonly known as "bay land" is one such micro-category and is characterized by open bush mixed vegetation, which is used for camel's and goats' browsing. This grazing management system is used when feed is scarce in the grass-dominated communal open grazing areas.

iii. Enclosures are areas that are closed off from grazing for a given period to foster vegetation regeneration [28]. These areas are often fenced using live fencing consisting of bushes; thus, they form an important component of rangeland landscape rehabilitation [29]. Enclosures are generally used for hay production, which is cut and carried to the livestock when there is a feed shortage for grazing in the open communal grazing areas. Pastoralists in the Somali region have started setting aside part of their rangelands physically or using social bylaws as fenced grazing reserves since the early periods of the second half of the twentieth century [30].

*2.3. Site Selection and Sampling Design*

The three aforementioned traditional rangeland management systems were selected for this study. Prior to the field layout and sampling techniques, a reconnaissance survey (focus group discussion) was conducted with elder pastoralists and resource managers who have indigenous knowledge of the sites. Information on the land-use condition and management was gathered, from past to present, using focus group discussions.

Transect survey methods were used for two traditional rangeland management systems: communal open grazing areas and bush-dominated browsing areas. For the communal open grazing type, a total of nine square plots of 400 m$^2$ each, were established at an interval of 5 km—heading from Harishin to Kebri Beyah rangelands of the Jigjiga zone (Figure 1). For the browsing management type, nine 400 m$^2$ square plots were also laid at an interval of 1 km from Awebere rangelands of Jigjiga zone (Figure 1). The size of the enclosure management practices in the study area was 1–2 ha, and the surroundings were fenced using thorn bushes. For sampling the vegetation and soil within these enclosures, three 400 m$^2$ plots were placed randomly within the three private enclosures aged 20, 25, and 30 years. In total, nine plots (three plots within every three enclosures) were selected.

In the selection of enclosure sites, we used a paired-site design, whereby the communal open grazing areas (used as a control site) were adjacent to selected enclosures. We assumed that both the enclosures and the communal open grazing areas had similar initial conditions at the time of enclosure establishment because the enclosures were established on some parts of the communal open grazing areas. This assumption is crucial to ensure that changes in carbon stocks are a result of the sole effect of the establishment of the enclosure (see [5]).

*2.4. Woody Vegetation Sampling*

All woody species encountered in each 400 m$^2$ plot were recorded. Woody vegetation structure was quantified by measuring: tree/shrub densities, canopy diameters, canopy heights, and stem heights of the woody species. Canopy cover was calculated using the average of the two longest canopy diameters perpendicular to each other and parallel

to the ground. Stem height was measured as the total height of the plant stem from the ground level to the highest foliage. For trees with multiple stems, each stem was measured separately, and the average was taken for the sample's stems. Height measurements and canopy lengths and widths were, however, conducted for the whole plant by measuring multiple stems (i.e., as if it was one tree). Trees with multiple stems at 1.3 m height were treated as a single individual with diameter at breast height (DBH) of the main stem as indicated by [31]. Tree/shrub aboveground biomass (AGB) was estimated using a nondestructive method by biomass regression equations (allometric equations) developed by [32] for the estimation of specific trees' or shrubs' carbon stock. Root biomass estimates were considered as 20% of aboveground biomass [33,34]. The total biomass was calculated by summing the above- and belowground biomass. The biomass was afterwards converted to carbon (C) by assuming a 50% biomass to carbon content [34].

### 2.5. Herbaceous Biomass and Soil Physicochemical Properties Sampling

For sampling herbs, biomass, and soil physicochemical properties, five subquadrats of 1 m$^2$ were laid out inside each 400m$^2$ larger plot (i.e., four at all corners and one at the middle position of each 400 m$^2$ plot; Figure 2), making a total of 135 plots. Herbaceous biomass measurements were taken during the flowering stages of most herbaceous species for ease of identification from September to December for all study plots. Aboveground herb mass was estimated by harvesting live and dead material at ground level from the 1 m$^2$ quadrats. The harvested samples were weighed immediately and ~30% were retained for determination of dry matter content (oven-drying at 105 °C for 48 h). The biomass was converted to carbon by assuming a 50% biomass to carbon content. Root carbon estimates were considered to be 20% of aboveground carbon [34].

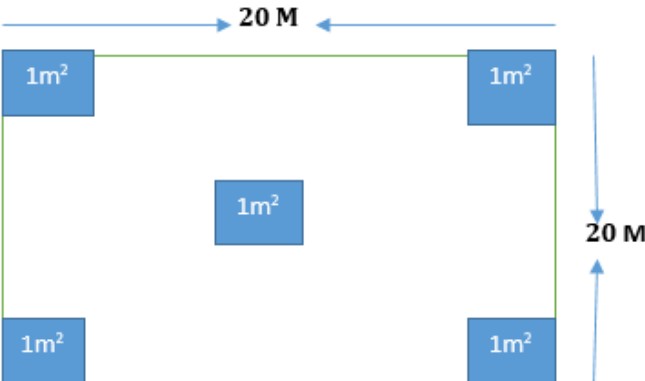

**Figure 2.** Field plot layout of the experimental site. Each 20 m × 20 m (400 m$^2$) plot was used for tree/shrub sampling, and the 1 m$^2$ quadrat nested within was used for sampling herb biomass and soil physicochemical properties.

Soil samples were taken at a depth of 30 cm using an auger, in each study plot and from the same five subquadrats used for herb biomass. Each set of five soil samples was mixed properly, air-dried, and passed through a 2 mm sieve to prepare it for analysis. Soil parameters were analyzed for soil texture (%silt, %clay, and %sand), bulk density, pH, and organic carbon (OC). Soil texture was determined using the hydrometer method [35], while bulk density was determined through the core method. A known volume of motorized soil coring was used for the determination of bulk density at a 30 cm depth level. Soil bulk densities were calculated as the ratio of the mass of the oven-dried soil sample to its core volume. We determined the soil pH by using a pH meter with 1:2.5 soils: water ratio (*w/v*), and the CEC by using the ammonia acetate method at pH 7. Nitrogen content was determined using Kjeldahl's method [36], and C: N ratio was computed. Walkley and Black's titration method was used to measure soil carbon concentration [37]. The percent soil organic matter was calculated by multiplying the percent organic carbon by a factor of 1.724 (https://www.agric.wa.gov.au/measuring-and-assessing-soils/what-soil-organic-

carbon; accessed on 10 December 2021). The EC was measured by conductivity meter using a 1:1 soil: water ratio. In all cases, standard chemicals and reagents were used for equipment calibration and standard curve plots, as required.

The content of organic carbon in soil estimated in percentage terms was converted to tons per hectare using bulk density, depth of the soil, and area (10,000 m$^2$) based on the procedure given by [38].

$$\text{Soil Organic Carbon (SOC; t/ha)} = \frac{\text{Soil mass in } 0 - 30 \text{ cm layer } \times \text{ SOC concentration (\%)}}{100} \tag{1}$$

$$\text{Soil mass} < 2 \text{ mm soil (t/ha)} = [\text{area (10,000 m}^2\text{/ha) depth (0.3 m)} \times \text{bulk density (t/m}^3)] \tag{2}$$

Information on soil texture, soil pH, nitrogen, organic carbon, bulk density, and vegetation across plots and management systems is summarized in Table 1.

**Table 1.** Soil and vegetation parameters under customary grazing management systems in eastern Ethiopia.

| Grazing Management Systems | Soil Parameters | | | | | Woody Vegetation | |
|---|---|---|---|---|---|---|---|
| | Soil Texture | Soil pH | N (%) | OC (%) | Bulk Density (g·cm$^{-3}$) | Tree Density (Stem·ha$^{-1}$) | Woody Canopy Cover (m$^2$/plot) |
| Enclosure | Loam | 8.61 | 16 | 32 | 1.29 | 50 | 38.54 |
| Browsing area (bay) | Loam | 7.55 | 19 | 22 | 2.18 | 1125 | 374.71 |
| Open grassland grazing area | Loam | 7.21 | 12 | 20 | 1.17 | 300 | 119.52 |

N = Nitrogen; OC = Organic carbon.

*2.6. Data Analysis*

All analyses were performed with the R Statistical Software version 4.1.1 [39]. In this study, carbon stock was assessed as the amount of carbon stored in various compartments (herbs aboveground and belowground, woody vegetation aboveground and belowground, and SOC). Thus, the carbon stock data from various compartments (herbs aboveground and belowground, woody vegetation aboveground and belowground, and SOC) were subjected to ANOVA using the "*aov*" function to determine the main effects of management practices on carbon stock. It is important to note that the analyses of woody vegetation aboveground and belowground carbon stock was based on the larger plots of 400 m$^2$ as the experimental unit, whereas the analyses of herbs aboveground/belowground and soil carbon were based on the smaller plots of 1 m$^2$. Means with significant differences among management practices were computed, and compared with the Student–Newman–Keuls post hoc test for differences in means, performed using the *SNK.test* function from the agricolae package (version 1.4.0).

We could not evaluate the carbon sequestration rate for the three management practices because we were not aware of the initial situations in the browsing bush areas. However, because enclosures were communal open grazing areas that have been enclosed for 20–30 years, we assessed the potential effect of enclosure on carbon stock. For each pair of communal open grazing-enclosure sites, we calculated the effect size as the natural log of response ratio [5,40], referred to as the restoration effect (RE):

$$\text{RE} = \ln\left(\overline{X1}\right) - \ln\left(\overline{X2}\right) \tag{3}$$

where RE is the log of the proportional difference between the groups [5,40]; (*X1*) is the carbon stock in soil, trees, and herbs in the enclosure; and (*X2*) is the carbon stock in the adjacent open grazing areas. We then computed the relative (i.e., to the initial value of

carbon represented by the carbon stock in open grazing areas) rate of change in carbon as follows:

$$\text{Relative rate of change in C} = \left( \frac{\text{LE}}{\Delta \text{ age}} \right). \tag{4}$$

where LE is the back-transformed log (RE) in %, i.e., $100 \times (\exp^{RE} - 1)$, and $\Delta$age is the number of years since the enclosure was established. Note that RE was back-transformed and expressed as percentage carbon stock change for better interpretation of the results. Trends in the relative rate of change in C with increasing enclosure age were analysed using scatterplots.

## 3. Results

### 3.1. Effect of Grazing Management on Carbon Stock

#### 3.1.1. Woody Carbon Stock

Aboveground woody carbon stocks were significantly (*ANOVA: F* = 13.625, df = 2, *p* < 0.0001; Table 2) higher in the browsing (bay) areas than in the rangeland units managed as enclosure and open communal grazing. Similarly, the belowground (root) woody carbon stocks were significantly different (*F* = 13.619, df = 2, *p* < 0.0001) among the three management practices, and highest for the browsing bay system (Table 2). However, the woody aboveground and root carbon stocks were not significantly different between enclosure and open communal grazing (Table 2).

**Table 2.** Variation in the carbon stocks (t C ha$^{-1}$) from the five compartments (above- and belowground herb, above- and belowground tree, and soil organic carbon (SOC)).

| Management Type | WAGC | WBGC | HAGC | HBGC | SOC |
|---|---|---|---|---|---|
| Enclosure | 1.34 [b] | 0.267 [b] | 2.00 [a] | 0.40 [a] | 85.11 [a] |
| Browsing land (bay) | 8.63 [a] | 1.73 [a] | 1.10 [b] | 0.22 [b] | 84.00 [a] |
| Open grazing land | 3.40 [b] | 0.68 [b] | 0.37 [b] | 0.06 [b] | 54.33 [b] |
| Mean | 4.46 | 0.89 | 1.16 | 0.23 | 74.48 |
| Df | 2 | 2 | 2 | 2 | 2 |
| *p* value | <0.001 | <0.001 | <0.001 | <0.001 | <0.001 |

WAGC = woody aboveground carbon; WBGC = woody belowground carbon; HAGC = herbs aboveground carbon; HBGC = herbs belowground carbon; means with same letter superscripts in columns were not significantly different at 0.05 level of significance.

#### 3.1.2. Herb Carbon Stock

Aboveground carbon stocks in herbs showed significantly (*F* = 9.994, df = 2, *p* < 0.0001) higher values in the enclosure than in the browsing and open communal grazing management systems. Similarly, the belowground (root) carbon stocks of herbs were significantly different (*F* = 10.151, df = 2, *p* < 0.0001) among the three management practices, with the enclosures recording the highest amounts of carbon. However, there was no significant difference of carbon between the browsing and open communal grazing management types (Table 2).

#### 3.1.3. Soil Organic Carbon Stock

Significantly lower (*F* = 20.644, df = 2, *p* < 0.0001) soil organic carbon was recorded in the communal open grazing system than in the other two customary rangeland management practices. However, the soil organic carbon stocks between the enclosure and open browsing management systems were not significantly different in the study area (Table 2).

### 3.2. Relative Rate of Carbon Change

The patterns in the relative rates of change in carbon, induced by the establishment of the enclosure, were positive (3% year $^{-1}$ for soil, 11.9% for wood, and 57.6% for herbs), indicating an increase in carbon stock following enclosure, as compared to the communal open lands. However, the patterns in the rates of increase in carbon with increasing age

varied with the compartments (Figure 3). For soil, the relative rate of change in carbon fluctuated with enclosure age, showing a hump-shaped trend (Figure 3), that is, an initial increase in soil C followed by a drop after 25 years of enclosure (Figure 3). Different patterns were observed for the herbs and wood, which showed an overall increase in C with age, and a remarkably sharp increase in old enclosures (Figure 3).

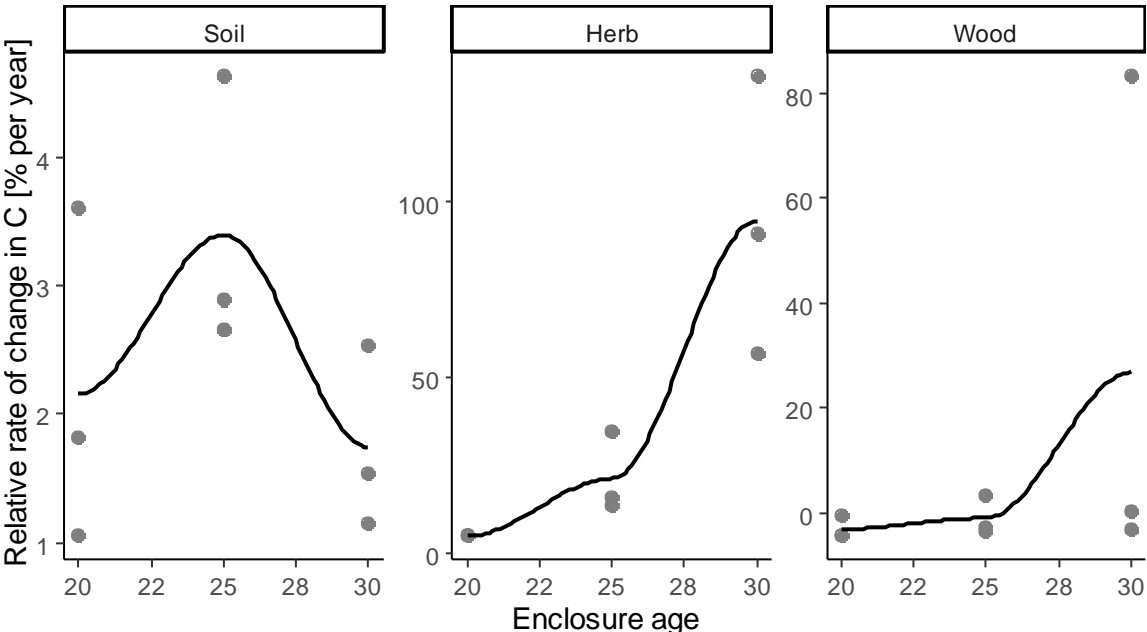

**Figure 3.** Trends in the relative rate of change in soil, herbs, and woody carbon stocks with increasing enclosure age.

## 4. Discussion

This study showed three important results. Firstly, contrary to our expectations, enclosures had a lower woody carbon storage, which was largely attributed to the role played by the aboveground woody plant coverage. Secondly, the browsing land and enclosure management types sequestered substantial amounts of SOC storage by differing only in terms of woody and herbs biomass. Thirdly, the establishment of enclosures induced an annual increase in carbon stocks of soil (3%), woody (11.9%), and herbaceous (57.6%) biomass, when compared to communal open lands. We discuss these findings below.

### 4.1. Effect of Customary Grazing Management on Woody Carbon Storage

Overall, we found that the above- and belowground woody carbon stocks were significantly different ($p < 0.001$) among the three management practices, with the lowest woody carbon stock and storage potential in the enclosure management practice. This difference in woody vegetation is due to the pastoralists' usual practice of frequently clearing most of the shrubs and trees from the enclosures and leaving a few species that can be used to provide shade for their animals. Our findings contrast with a previous study in the southern part of Ethiopia by [41], who reported higher woody carbon storage inside enclosure management areas as compared with open communal grazing management areas. The clearing of shrubs and trees inside enclosure lands used to produce grasslands has become a common management strategy by some Somali pastoralists in eastern Ethiopia [24]. Consequently, this led to the lower woody vegetation carbon stock for the enclosure management system in the study area.

By contrast, we found the highest above- and belowground woody carbon storage for the browsing land management type. Many studies [17,41,42] noted that the process of open grassland savannas being transformed into thick bushes seems to be beneficial for maintaining carbon stocks or reducing their losses, thereby contributing to climate

change mitigation in semiarid rangelands. In line with this, the browsing management practice that dominated woody plant species in the study area had an increased potential for woody carbon stock. If grazing on a particular ecological site influences the abundance of woody plants, the effect on carbon dynamics can be large [43]. Hence, the growth of woody vegetation makes it possible to store carbon in woody material that persists longer than herbaceous matter [44].

### 4.2. Effect of Grazing Management on Herbs Carbon Storage

A significant amount of aboveground and belowground herbs carbon storage was noted for the enclosure management type when compared to the other two management practices. This might be attributed to increased herbs biomass and cover through the exclusion of grazers. Interestingly, the bush-dominated land had a major negative influence on herbaceous material aboveground and belowground carbon storage, with the lowest carbon storage found for the browsing land management practice. This finding is in line with previous studies [13], which reported lower herb carbon storage potential in bush-dominated lands when compared with communal open grazing lands (in Borana rangelands in Southern Ethiopia).

### 4.3. Effect of Grazing Management on SOC Carbon Storage

The SOC stocks were significantly different ($p < 0.001$) among the three management practices. The SOC stock for the communal open grazing land practice was significantly lower by 30 t C ha$^{-1}$ than the enclosure and browsing land management practices. In open-access grazing systems, where mobile and sedentary forms of livestock husbandry coexist, rangelands are exploited by multispecies herds, and grazing can cause defoliation of plants [45,46]. In this context, livestock grazing intensity is thought to have a major impact on soil carbon storage in rangeland ecosystems [10,11,47,48]. Moreover, trampling, as one component of grazing, has negative effects as it accelerates the deterioration of vegetation, transforming standing materials into litter and incorporating litter into the soil [45]. On different soil types, trampling breaks surface crusts, affects water infiltration, compacts the soil, and reduces infiltration in clay [48]. Consequently, excessive trampling reduces nutrient contents, such as nitrogen in soil, and can further limit plant growth. Over time, this process may exhaust carbon stock and the capacity of the open grazing area to store carbon [49]. Our findings showed that the nitrogen content in the open grazing management system area was lower than that in the browsing and enclosure areas (Table 1). This could be attributed to an existing lower carbon stock in the open grazing areas in addition to continuous heavy-grazing pressures.

Our results also showed that, from the perspective of SOC storage, the enclosure and browsing management practices would contribute more positively to mitigating climate change in the study area. The highest SOC storage was recorded for the enclosure management practice in the study area. Similar findings were reported in other studies, which found increased levels in soil C stock storage in enclosure land-use practices [10,50,51]. A study conducted in the southern parts of Ethiopia reported that the conversion of the rangeland ecosystem from communally owned grazed areas to grazing enclosures resulted in a 29.98% increase in SOC sequestration and an increase of 68.36% for vegetation carbon sequestration [52]. Minimizing soil disturbance and leaving adequate residue are also likely to prove useful in retaining carbon in rangelands [43]. In our study areas, the highest woody biomass was found for the browsing management system, and the highest herb biomass was found in the enclosure area. This contributed to persistent larger belowground carbon stocks for the browsing and enclosure management systems. Organic matter inputs to the soil come mainly from trees (e.g., through litter, fine roots, and exudates) and from herbaceous vegetation [5,40]. Moreover, many studies reported that about 90% of carbon in rangelands is stored in the form of soil organic carbon [53]. Importantly, for the rangeland ecosystem in our study area, approximately 93% of the carbon was stored in the form of SOC. Therefore, management practices that increase organic matter inputs to soils or

decrease losses from soil respiration and erosion can sequester additional carbon, while actions that decrease carbon inputs or increase losses should be avoided [54].

*4.4. Relative Annual Rate of Carbon Change*

The analyses of the relative rate of change in carbon revealed an annual increase of 3%, 12%, and 56.7% in the soil, woody vegetation, and herbs, respectively, within the enclosure system. These results collectively suggested that enclosures have a great potential to sequester organic carbon in a semiarid environment. In particular, the 3% annual increase in the enclosure soil, as compared to that of the communal grazing land used as a reference, suggested that the latter stored increasing amounts of C in the initial years after its establishment. This finding further demonstrates that enclosures also have the potential to sequester soil carbon at rates greater than the annual increase rate of 4‰ per year$^{-1}$ recommended by the "4 per 1000 Soils for Food Security and Climate" initiative (https://www.4p1000.org/; accessed on 28 March 2022). However, the annual rate of carbon change did not increase indefinitely with enclosure age (Figure 3) but declined after 25 years. The resulting hump-shaped trend in the relative rate of change in carbon in the enclosure soil corroborates well with the "sink saturation" hypothesis. The sink saturation hypothesis posits that sequestration rates increase in the initial years but decline as time progresses and soils approach a new equilibrium [55,56]. It is, therefore, plausible that enclosure potential to store soil carbon is time-bounded, as reported in agricultural landscapes in northern Ethiopia [5]. As opposed to the trends for the soil component, the annual rate of carbon change increased drastically in the woody and herbs components after enclosure establishment. The increase in both the herbs and woody components was attributed to limited grazing disturbances, which contributes to restoring plant growing conditions in enclosures. This finding depicts a possible trade-off pattern mechanism in the carbon sequestration and allocation among soil, herbs, and woody vegetation, as recently observed in tropical montane forests [40].

**5. Conclusions**

Pastoralists in the Somali region have various traditional natural resource management strategies using ecosystem and herd composition indigenous knowledge. In this study, we measured the carbon stock of woody vegetation, herbs, and SOC in semiarid pastoral ecosystems under three different rangeland management practices. We found 95, 89, and 47 t C ha$^{-1}$ total storage potential (aboveground and belowground) for the open browsing, enclosure, and open communal grazing areas, respectively. This result led us to two critical implications. Firstly, the performance and the ability of a rangeland to serve as a carbon storage will depend on the overall health of the rangeland. Accordingly, the stress resulting from a high grazing pressure throughout the year on open grazing areas is generally linked to diminished woody and herb biomass. This is attributed to a reduction in soil, woody vegetation, and herbs carbon storage in the open grazing lands. Secondly, the modest changes in C storage in rangeland ecosystems have the potential to modify the global C cycle and indirectly influence climate mitigation. Thus, the change from open grazing land to enclosure and browsing management practices resulted in the highest levels of carbon storage. The establishment of enclosures increased carbon storage at a rate of 3, 11.9, and 57.6 percent for the soil, woody vegetation, and herbs, respectively, as compared to communal open lands. This study indicates the importance of pastoral ecosystems in mitigating the release of $CO_2$ and influencing climate change and global warming. This is a great opportunity for poor nations, such as Ethiopia, to contribute to mitigating climate change. Moreover, the soil carbon stock had a 93% proportion, meaning that any alternative land-use to pastoralism that exposes soil carbon would have substantial adverse environmental effects. Yet, the current study was only focused on three grazing management practices; there is a need for further identification and investigation of various customary natural resource management strategies and their co-benefits for climate change mitigations.

**Author Contributions:** Conceptualization, H.H.G.; methodology, H.H.G.; software, H.H.G.; validation, H.H.G.; formal analysis, H.H.G.; investigation, H.H.G.; resources, H.H.G., S.W.D. and A.E.; data curation, H.H.G.; writing—original draft preparation, H.H.G.; writing—review and editing, S.M., A.J.M.D., A.E. and S.T.; visualization, H.H.G.; supervision, T.Z.K., Y.T., S.T., S.W.D. and A.E.; project administration, H.H.G., S.W.D. and A.E.; funding acquisition, H.H.G., S.W.D. and AE. All authors have read and agreed to the published version of the manuscript.

**Funding:** This study was financially supported by: (1) Africa Center of Excellence for Climate Smart Agriculture and Biodiversity Conservation (ACE Climate SABC) of Haramaya University; (2) the UK Research and Innovation (UKRI) through the Global Challenges Research Fund (GCRF) program, grant ref: ES/P011306, implemented by the Regional Universities Forum for Capacity Building in Agriculture (RUFORUM); (3) the BAYER foundation, with the support of the Alexander von Humboldt Foundation (AvH), through the program "AGNES-BAYER Science Foundation Research Grant for Biodiversity Conservation and Sustainable Agriculture in Sub-Saharan Africa".

**Institutional Review Board Statement:** Not applicable.

**Informed Consent Statement:** Not applicable.

**Data Availability Statement:** Data are available upon request from the authors.

**Acknowledgments:** This research was supported by the CGIAR Research Program on Climate Change, Agriculture and Food Security (CCAFS) and the Global Research Alliance on Agricultural Greenhouse Gases (GRA) through their CLIFF-GRADS program. CCAFS capability building objectives were carried out with support from CGIAR Trust Fund and through bilateral funding agreements. For details, please visit https://ccafs.cgiar.org/donors (accessed on 17 April 2022). We thank the CIRAD (French Agricultural Research Centre for International Development), Senegal Office, Dakar, for hosting the recipient, and the government of New Zealand for providing financial support. The first author, HHG, would also like to acknowledge Tim Newbold and the reviewers for their comments, which helped improve the manuscript significantly.

**Conflicts of Interest:** The authors declare no conflict of interest.

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
