# Peer review of "Carbon Stock and Change Rate under Different Grazing Management Practices in Semiarid Pastoral Ecosystem of Eastern Ethiopia"

_land, doi:10.3390/land11050639_

Round 1

Reviewer 1 Report

Introduction

Line 73: also soil properties (e.g. clay content, amorphous Fe- and Al-oxides)

Material and Methods

Are the plots representative of the study area and the management type (vegetation, soil) ?

Information concerning bedrock and soil types in the study areas are missing. 

Information concerning soil type and basic soil properties (e.g. soil depth, soil water regime,  humus content, soil texture, amorphous Fe- and Al-oxides, pH) in the plots are missing.

Unfortunately, root biomass was not determined (roots are the most important C-input into the soil)

Information about the rooting depth and vertical root distribution would be desirable. 

Line 224-228: Are the contents of gravel and rocks considered in the calculation? What was their proportion?

Results

A table containing information about organic carbon concentrations and carbon stocks in soil and vegetation for the different management types are needed.

Discussion

Line 345-346:  excessive trampling reduces the nutrients content like nitrogen on soil: an explanation is necessary !

Line 374-375: an annual increase by 3 % in the soil is extremely high.

Line 381: the annual rate is 0,4 % per year

Conclusions

Line 411-413: What does this mean for Ethiopia concerning carbon sequestration and climate change?

Author Response

Reviewer #1:

Introduction

Comment 1. Line 73: also soil properties (e.g. clay content, amorphous Fe- and Al-oxides)

Response to comment 1. “soil properties (e.g. clay content, amorphous Fe- and Al-oxides)” have been also added to the factors, as suggested by the reviewer. Please see lines 73-74

Material and Methods

Comment 2. Are the plots representative of the study area and the management type (vegetation, soil)?

Response to comment 2. The plots were established using the transect method to cover not only a wider range of the environment in the study area but also the management type (please see lines 160-169). In addition, a plot size of 400m2 in the grazing land ecosystems would be enough to capture most of the variability in the vegetation. For the herb biomass and soil sampling, a total of 5x27 = 135 plots of 1m2 each were used (see lines 199-201; and Fig 2).

Comment 3. Information concerning bedrock and soil types in the study areas are missing. 

Response to comment 3. The Information concerning bedrock and soil types has now been added. Please see lines 107-110.

Comment 4. Information concerning soil type and basic soil properties (e.g. soil depth, soil water regime, humus content, soil texture, amorphous Fe- and Al-oxides, pH) in the plots are missing.

Response to comment 4. We agree with the reviewer that some information on soil properties was missing. We have provided in Table 1. Information on soil texture, soil pH, Nitrogen, Organic Carbon and bulk density. Please see Table 1

Comment 5. Unfortunately, root biomass was not determined (roots are the most important C-input into the soil).

Response to comment 5. Yes, we fully agree with the reviewer that root biomass was not measured in the field, mainly because of the limitation of resources. But it was estimated from aboveground biomass (see lines 191-193). See also Table 2 which provides the summary of woods and herbs belowground carbon.

Comment 6. Information about the rooting depth and vertical root distribution would be desirable. 

Response to comment 6. Yes, but unfortunately, we could not get the rooting depth and vertical root distribution information in our study plots.

Comment 7. Line 224-228: Are the contents of gravel and rocks considered in the calculation? What was their proportion?

Response to comment 7. We did not consider gravel and rock mass and we correct the formula by taking the soil Mass < 2mm soil (ha). 

Results

Comment 8. A table containing information about organic carbon concentrations and carbon stocks in soil and vegetation for the different management types are needed.

Response to comment 8. During the revision, we have added Table 1 which now provides information on organic carbon concentrations in the soil, and vegetation parameters for the different management systems. In addition, the Table 1 provides also information on soil texture, pH, bulk density and nitrogen content for the different management types. Finally, Table 2 in the revised paper (initially Table 1 in the first version) captures information on carbon stocks in aboveground and belowground wood and herb, and soil compartments (see Table 2).

Discussion

Comment 9. Line 345-346:  excessive trampling reduces the nutrients content like nitrogen on soil: an explanation is necessary !

Response to comment 9. We have now elaborated on the effect of excessive trampling. Please see lines 359-363.

Comment 10. Line 374-375: an annual increase by 3 % in the soil is extremely high.

Response to comment 10. The enclosures in our study area were established in the heavy degraded open grazing communal land, which had very low initial SOC. As we observed, the mean of SOC in the enclosures was twice greater than the open grazing communal land. As a result, this may contribute to having a high uptake rate of SOC and an extremely increase annual sequestration rate in the enclosures. Similar high rates have also been reported in previous studies (e.g. 7–18.9% per year; see Noulèkoun et al. 2021 Grazing exclosures increase soil organic carbon stock at a rate greater than “4 per 1000” per year across agricultural landscapes in Northern Ethiopia. Sci. Tot. Envir. https://doi.org/10.1016/j.scitotenv.2021.146821).

Comment 11. Line 381: the annual rate is 0.4 % per year

Response to comment 11. True. This has been corrected. Please see line 398.

Conclusions

Comment 12. Line 411-413: What does this mean for Ethiopia concerning carbon sequestration and climate change?

Response to comment 12. We provided information as our result indicating the importance of pastoral ecosystems for absorbing a significant amount of carbon from the atmosphere as a means of mitigating climate change. Please see lines 434-435

Reviewer 2 Report

Peer review of Land manuscript 1686855

Title: Carbon stock and change rate under differentiated grazing management practices in semiarid pastoral ecosystem of eastern Ethiopia

Authors: H.H. Gebremedhn, T.Z. Kelkay, Y. Tesfay, S. Tuffa, D.W. Sintayehu, S. Mensah, A.J.M. Devenish, A. Egeru.

Overall, I found the manuscript well organized and prepared. Somewhat troubling was learning that several references were not as described, particularly for soil organic carbon. The study would have been much stronger if soil organic carbon had been sampled and not predicted using a conversion factor because it is important in the ecosystems studied. Most of the English language in the manuscript is adequate, although improvements can be made in several places.

Specific comments:

Title: Consider if “ecosystem” should be plural as “ecosystems.” At lines 128-129 you refer to dividing grazing habitat into categories based on different units of land defined by plant cover, soil type and ecosystem functioning, which (to me) suggests several ecosystems of differing scale.

Line 166: Were control sites selected randomly in relation to the enclosure sites or based on certain criteria? More description of control site selection is needed.

Line 204: Please confirm that root carbon predictions are provided in reference [32] as 20% of above ground biomass; I did not see that information in the reference provided.

Line 221: Reference [37] relates to alkaline materials added to mine spoils, not soil carbon. The conversion factor 1.724 is not provided in the reference.

Line 226 Has SOC been defined previously? I presume the acronym stands for soil organic carbon. “OC” is identified at line 213, but not SOC.

Line 228: For the data analysis, define the experimental unit.

Line 229: RStudio Team 2020 should have a reference number, not a year.

Line 236: Identify the method used for separation of management type means, such as Tukey's range test.

Line 268: In the caption for Table 1, it would be helpful to define SOC.

Line 518: https://doi.org/10.4337/9781784710644 is broken or not active; please correct.

Line 526: The link to https://doi.org/10.21000/JASMR90010227 is broken.

Author Response

Reviewer #2:

General comment. Overall, I found the manuscript well organized and prepared. Somewhat troubling was learning that several references were not as described, particularly for soil organic carbon. The study would have been much stronger if soil organic carbon had been sampled and not predicted using a conversion factor because it is important in the ecosystems studied. Most of the English language in the manuscript is adequate, although improvements can be made in several places.

Response to general comment. We are grateful for the time and energy the reviewer put into reviewing this manuscript. We further really appreciate the reviewer’s positive feedback on the content.

Specific comments:

Comment 1.

Title: Consider if “ecosystem” should be plural as “ecosystems.” At lines 128-129 you refer to dividing grazing habitat into categories based on different units of land defined by plant cover, soil type and ecosystem functioning, which (to me) suggests several ecosystems of differing scale.

Response to comment 1. In this study, we considered pastoral as one broad ecosystem type. The dividing grazing habitat into different categories is within that one broad pastoral ecosystem. So, we would prefer to keep it singular since we already referred to the Differentiated Grazing Management Practices in the title.

Comment 2. Line 166: Were control sites selected randomly in relation to the enclosure sites or based on certain criteria? More description of control site selection is needed.

Response to comment 2. The control sites are the communal open grazing plots which were established at an interval of 5 km along the transect (see lines 160-163). We called them control because the subsequent enclosure sites used in this study were adjacent to these communal open grazing/control sites. Please see also lines 170-175.

Comment 3. Line 204: Please confirm that root carbon predictions are provided in reference [32] as 20% of above ground biomass; I did not see that information in the reference provided.

Response to comment 3. We have now corrected this, and it is referenced [33, 34] in the revised version. It has been specified in line 192.

Comment 4. Line 221: Reference [37] relates to alkaline materials added to mine spoils, not soil carbon. The conversion factor 1.724 is not provided in the reference.

Response to comment 4. We thank the reviewer for the correction. We have deleted reference (37) and replaced it with https://www.agric.wa.gov.au/measuring-and-assessing-soils/what-soil-organic-carbon

Comment 5. Line 226 Has SOC been defined previously? I presume the acronym stands for soil organic carbon. “OC” is identified at line 213, but not SOC.

Response to comment 5. SOC has now been defined in the equation. Please see line 230.

Comment 6. Line 228: For the data analysis, define the experimental unit.

Response to comment 6. The experimental unit was the bigger plots of 400m2 for the woods above-ground carbon and woods belowground carbon, and the smaller plots of 1m2 for the herbs and soil carbon. This has now been specified. See lines 243-246

Comment 7. Line 229: RStudio Team 2020 should have a reference number, not a year.

Response to comment 7. This has now been provided as a Reference [39]

Comment 8. Line 236: Identify the method used for separation of management type means, such as Tukey's range test.

Response to comment 8. This has now been clarified in Lines 249-252

Comment 9. Line 268: In the caption for Table 1, it would be helpful to define SOC.

Response to comment 9. SOC has now been defined in the caption for Table 2, initially Table 1 in the first submission. See line 284

Comment 10 Line 518: https://doi.org/10.4337/9781784710644 is broken or not active; please correct.

Response to comment 10. We have now corrected this. Please see below:

Odoemene A. Climate change and land grabbing. In Angelo MJ and Du Plesis A. Research Handbook on Climate Change and Agricultural Law, pp 423–449. 2017. https://doi.org/10.4337/9781784710644

Comment 11. Line 526: The link to https://doi.org/10.21000/JASMR90010227 is broken

Response to comment 11. The link was in the reference [37], which has been deleted during our revision, referring to reviewer comment 4 above.

Round 2

Reviewer 1 Report

The manuscript should be accepted.